# Innovative Arrangements between Public and Private Actors in Affordable Housing Provision: Examples from Austria, England and Italy

**Gerard van Bortel * and Vincent Gruis**

Department of Management in the Built Environment, Delft University of Technology,
2628 CE Delft, The Netherlands; v.h.gruis@tudelft.nl
*   Correspondence: g.a.vanbortel@tudelft.nl

**Abstract:** Affordable housing is increasingly developed, financed and managed by a mix of state, third-sector, market and community actors. This has led to the emergence of various hybrid governance and finance arrangements. This development can be seen as part of a general long-term neoliberal trend in government policies, and social, cultural and economic developments. It is therefore likely that the hybridity and variety of governance and finance of affordable housing will continue to grow. This article discusses innovative hybrid arrangements from Austria, England and Italy, in which governments, private and non-profit actors collaborate to increase the supply of affordable housing. These cases illustrate how the provision of affordable housing in a neoliberal context can benefit from the involvement of market actors and communities. Nevertheless, they also show that governments continue to play a crucial role in initiating and facilitating these arrangements.

**Keywords:** affordable housing; governance; co-production; non-for-profit housing; finance models; innovations

## 1. Introduction

Traditionally, social or public housing has been provided through a variety of actors, such as public agencies (state and local authorities), third-sector organizations (housing associations and foundations), and cooperatives or private investors (supported with state grants). The dominant type of provider and tenure (rent, ownership, cooperative) varies from country to country. Van Bortel, Gruis, Nieuwenhuijzen and Pluijmers conclude that "in recent decades, however, more hybrid shapes of provision have emerged, in which housing is delivered through cooperation between different types of actors, including a growing collaboration between tenants and professional housing providers, as well as an increasing mix of public and private finance" ([1] p. 2).

The trend towards more hybridic forms of housing provision can be explained by various developments. In general, it can be seen as a response to, or part of, a long-term neoliberal trend, and an associated retrenchment of the state in direct support and provision of social housing. This, for example, forces third-sector housing providers to rely more and more on private finance [2]. In the UK, this is visible in the housing associations' interest in attracting finance from institutional investors through bonds [3]. In France, tax incentives have led to an increased activity of private developers in affordable housing [4]. The abolishment of the public status of many social housing providers in Germany the 1980s, and the subsequent acquisitions of social housing portfolios by institutional investors (e.g., private equity funds, pension funds, insurance companies) can be seen as a "radical" front-runner in this respect. Thus, reduced state support, combined with the recent stagnating economy and often long waiting lists for existing social housing stock, have also stimulated citizens' interest in

becoming more active in providing their own housing. These collaborative arrangements of groups of citizens are often supported by public, third-sector and/or private actors [5]. The latter can also be seen as part of a wider trend of citizens wanting to collectively organize services. This development does not only take place in the housing sector, but in various other service domains, such as energy, child-care and education [6,7]. The increased involvement of citizens taking matters into their own hands is also embraced by national governments, with the Big Society and Localism agenda in the UK and the Participation Society in the Netherlands as notable examples [8].

The active role of citizens in the development of joint solutions to social problems and the provision of public services is often referred to as "co-production" or "co-creation". These terms are used as an umbrella concept encompassing a broad array of initiatives oriented towards the active involvement of users and collaboration between residents [9]. Many potential positive effects are accredited to increased resident involvement in housing provision, which can, for example, lead to higher housing satisfaction and increased social cohesion, as well as generating housing solutions that do not exist within the mainstream housing provision in terms of affordability, sustainability and lifestyle.

Several studies suggest that citizens' involvement can improve the efficiency and effectiveness of public service delivery, as well as strengthen the affective connection between citizens and government [10–13]. Other findings indicate that third-sector organizations are, in comparison to public and for-profit providers, better able to develop higher and more-sustainable levels of participation by citizens in public service delivery [9,14–16].

Market actors are increasingly involved in the provision of affordable rental housing. Multiple drivers can lead them to provide rental housing below full market rents. Sometimes market actors are subsidized—or otherwise incentivized—to provide housing below market rents. The substantial finance currently available to institutional investors has also increased interest among private investors to gear more of their investments towards affordable housing (e.g., institutional investments in affordable housing facilitated through tax incentives in France [4], and the acquisition of portfolios owned by Dutch housing associations by German (Patrizia) and UK (Roundhill) actors. Bonds constitute another form of private-sector investment in affordable housing. Especially in the UK, bond finance helps a growing number of housing associations to invest in affordable housing and balance the opportunities and risks in combining social and economic goals [17].

A potential benefit of private-sector involvement is the financial leverage that can be attained by combining public/third-sector and institutional finance. This can lead to increased investment capabilities and, perhaps, increased business efficiency if traditional social and public housing providers adopt more-businesslike behavior. This development is further enforced as a result of closer scrutiny by (commercial) lenders on business models, financial viability, management capabilities and transparency. Potential risks could relate to the question of how long-term market actors are committed to the affordable rental market, as well as how quality and sustainability of housing will be safeguarded in the context of the increasingly financial return-driven behavior of the actors involved.

This article discusses three innovative, hybrid arrangements in which governments have stimulated the involvement of private (market and community) actors in the provision of affordable housing, from Austria, England and Italy. These cases do not only have in common that they concern new forms of partnerships between public and private actors, they are also innovative through their governance arrangements, trying to mitigate some of the risks of public–private partnerships, as highlighted above (e.g., the short-term, exclusive and often "one-off" natures of some past initiatives). Moreover, these innovations are not only meant for the traditional target groups of social and public housing, but also for (lower-)middle income households that have difficulties in accessing affordable housing (see Table 1).

**Table 1.** Overview of the three case studies.

| | Austria | Italy | England |
|---|---|---|---|
| | **Vienna Housing Construction Initiative** *"Wohnbauinitiative"* | **System of Integrated Investment Funds** *"Sistema Integrato dei Fondi"* | **Limited Liability Housing Partnership** |
| **Affordability:** target group | Middle-income households | Lower-middle income households | Low-income households |
| **Governance:** allocations, rent setting and management arrangements | Limited-term restrictions on rents and allocations. Management by private investors, allocations during duration of the loan term conducted by the general allocation agency for subsidized housing. | Rent setting and allocation based on project-level agreement with local authority. Includes mechanisms to stimulate resident participation in design and management phases. | Allocation based on local authority waiting list. Rent setting based on income and National Living Wage criteria. Management through Limited Liability Partnership, sharing costs, risks and revenues. |
| **Finance:** investment sources | Local authority (cheap loans and/or land) in combination with private finance | Through national and regional funds sourced by a mix of public and private finance | Mix of local authority and housing association funding |
| **Hybridity:** partner mix | Local authority and profit and not-for-profit housing investors | Complex partnership of government, market, not-for-profit actors and residents | Local authority and private not-for-profit housing investor |

These case studies are taken from a larger study, encompassing 13 case studies from 10 countries, in which a group of researchers and practitioners explored innovations in the governance and finance of affordable housing in Europe and beyond [1]. Researchers from the European Network of Housing Researchers (ENHR) linked up with practitioners from the European Federation for Living (EFL) to describe thematic developments in various (mainly northwestern European) countries, illustrated with case studies from contemporary practice. The three case studies were selected because they represent different strategies by governments to include private and not-for-profit actors in long-term affordable housing provision. The selected case studies each focus on one specific local strategy and do not pretend to present a comprehensive overview of all national affordable housing strategies, hybrid or otherwise.

A generic framework was used for the analysis of all case studies. This framework included identical key questions and compulsory structural elements related to the purpose of the study. The research activities were executed during the period 2016–2017. No coding of documents or interview transcripts was used. Table 2 presents more in-depth information on the research methodology applied in each of the case studies.

**Table 2.** Overview of the three case study methodologies.

| Austria | Italy | England |
|---|---|---|
| The Viennese contribution builds on the analyses of available data and policy documents. Six interviews with policy practitioners and builders were conducted to retrieve important information on the functioning and organization of the Wohnbauinitiative in Vienna. The interview partners were also asked about their views on the pros and cons of the innovation. | The Italian contribution is based on practices by two case study authors working for the Fondazione Housing Sociale (FHS). The FHS is one of the key actors in the affordable housing innovation analyzed in this study. The third author originated from academia and ensured a critical reflection on the findings as well as incorporating reviews of policy documents and academic literature. | The UK contribution is based on a review of policy documents and academic literature. |

Following from the general discussion above, the discussion of these cases focuses on two main themes:

- How are private actors encouraged to participate in the provision and finance of affordable housing?
- What arrangements have been made to secure public housing values, such as quality, affordability and allocation to specified target groups?

The article starts with a defining the key concepts: affordable housing, housing governance, housing finance and hybridity. It continues with a description of the three case studies. First, we present the Wohnbauinitiative (WBI) in Vienna, which employs an innovative financing method for affordable housing, a method that simultaneously stimulates co-production relations between private, government and not-for-profit actors in the development of affordable housing. Second, we introduce the case of social housing in Italy, where new financing methods are combined with cohousing and collaborative housing characteristics. Third, we conclude this section with a discussion on collaboration between a housing association and a local authority in England that, again, combines co-production and an innovate financing method in delivering affordable housing. The article concludes with a general discussion of the findings following the two main themes.

## 2. Key Concepts

In this section, we discuss the key concepts in the article, building on definitions of affordable housing, housing governance, housing finance and hybridity derived from literature on housing studies.

### 2.1. Affordable Housing

Lack of affordable rental housing is mainly an urban issue, and is increasingly recognized as one of the major challenges facing European cities [18,19]. Many cities lack an adequate supply of affordable rental housing for low- and middle-income households, including for key urban professionals such as nurses and teachers. This situation may lead to soaring rents, high levels of commuting, congestion, poor tenure mobility and, ultimately, to cities where people work, play and shop, but do not live. In the current context of reduced job security and tougher lending conditions, the need for affordable housing is growing. The economic significance of well-developed affordable housing sectors is paramount. Only with sufficient and affordable housing supply can urban areas attract a qualified workforce at competitive personnel costs. Guaranteeing affordable housing is therefore one of the basic requirements for the development of opportunities and talent [20].

Before we can define "affordable housing", we first need to provide a workable description of "social housing". There is, however, no common definition of "social housing" at the European level. Social housing in the European context is characterized by a wide variety of country-specific systems and policies. Overall, the terms "social", "housing" and other related concepts are used in varying combinations in different Member States to refer to housing that is provided at below-market price for selected groups within the population. The latter are defined nationally, regionally or even locally, based on specific (housing) needs and eligibility criteria set by the relevant authority.

"Affordable housing" is often seen as a form of tenure that is complementary to social housing; however, for this concept, a clear cross-national definition is lacking [19,21]. Housing markets are nationally and locally specific. While definitions of affordability vary, it may be understood as housing that is adequate in standard and location for low- to middle-income households at fees that enable them to meet other basic needs on a sustainable basis [22]. Thus, the concept of affordable housing refers to housing for a broader range of household incomes than social housing. The affordable segment includes the gap between the traditional social and public housing segments and the level of expenditure that is still affordable for moderate-income households (based on housing expenses/income ratios or residual income) or that are not able to buy a home and cannot afford to pay full market rents (see Figure 1).

The size of the affordable housing segment can vary as a result of local market conditions. Rents in social housing (provided by not-for-profit organizations) and public housing (delivered by government

entities) are mostly determined related to costs or household income, and allocated based on needs or position on waiting lists. Affordable housing rents are derived from, but lower than, market fees. This segment borders the full market housing segment, and can be provided by various profit, non-profit and public actors. In Figure 1 we have incorporated two definitions of affordable housing. The narrow definition of affordable housing refers to housing for moderate-income households not eligible for social housing, but who are also not able to pay full market prices. The wider definition is often used as a concept to indicate the challenge of providing affordable housing in general, rather than for specific rental housing segments.

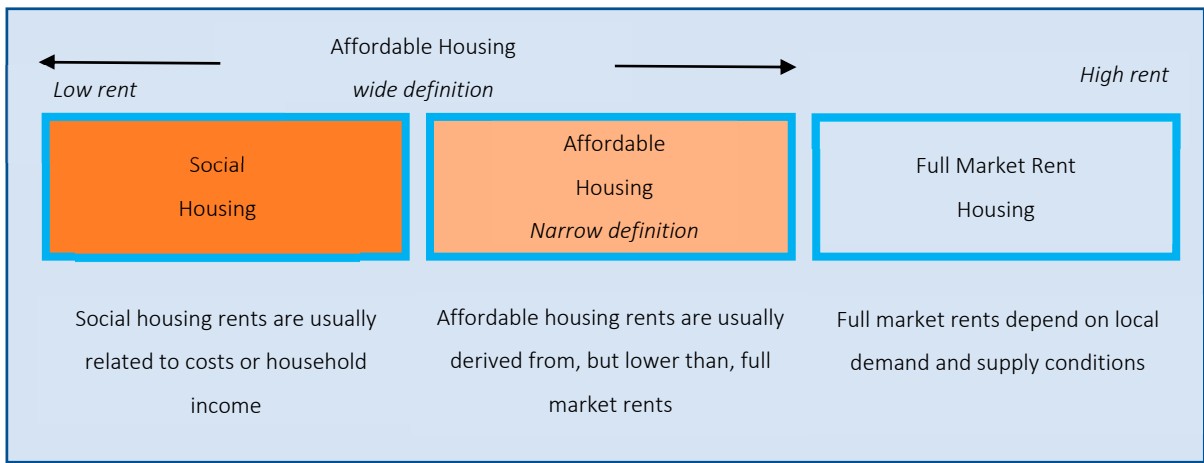

**Figure 1.** Rental housing segments. Source: [21], adapted by authors.

*2.2. Housing Governance*

Governance refers to the processes of governing undertaken by a government, market or network over a family, tribe, formal or informal organization or territory through laws, norms, power or language [23]. For the purpose of this article, we focus on governing processes between public, private and not-for-profit organizations. These processes take place within the context of national policies, economies and welfare regimes. Recent developments across Europe suggest a move towards welfare regimes with a more modern–corporatist character and an indirect style of governance [24]. This gives local authorities and non-state providers of welfare services, such as profit and not-for-profit actors, more freedom and autonomy in the delivery of public services ([1], p. 5).

There is a broad consensus that decision-making between actors has become more interdependent. There is, however, very little common ground among scholars and practitioners about the distribution of power and influence. Van Bortel et al. ([1], p. 5) contend that these "shifts in governance do not necessarily lead to a reduction of state power, but could indicate a shift from formal to informal techniques of government steering [25], such as steering in the "shadow of hierarchy" ([26], pp. 145–147), [27,28]. Rhodes [29] claims that interdependent networks of state and non-state actors weaken the hierarchical powers of the state. Davies [30,31], on the other hand, insists that the state is still dominant, and that power relations are asymmetrical and still favor the state. Similarly, Jones and Evans [32] contend that many fail to see the state-centeredness in many network arrangements. In several European countries, there is a clear movement of governments strengthening their influence on (social) housing providers by restricting activities and regulating their organizational characteristics (see [33] for examples of developments in the Netherlands, Sweden, France and Belgium (Flanders)).

Facing budget cuts and EU-imposed constraints on the state support of affordable rental housing, governments are increasingly looking to private-sector actors to expand housing supply [34]. At the same time, existing social landlords, forced to work without subsidies, are becoming more commercially oriented, and are increasingly adopting private-sector management strategies [35–38]. The inherent contradictions between affordability and the profit-maximizing nature of some actors generate new

governance challenges. It is within this complex landscape that we want to find innovative governance frameworks that combine the formal and informal, hierarchical, network and market-based instruments needed to support decision-making processes in the affordable housing sector [13].

### 2.3. Housing Finance

Basically, housing finance can be defined as the way in which cash is provided to develop and maintain affordable housing. Social housing is traditionally financed by mixing various sources. The most common combinations in northwestern Europe are:

- state guarantees, grants and/or loans combined with bank loans, attracted by the social housing providers and paid from rental income, in the case of social rent (public or third-sector);
- state grants and institutional investments, in the case of market actors providing social housing;
- state grants combined with mortgages, attracted by the household, in the case of home-ownership;
- state grants combined with cooperative funding mechanisms, in which households buy shares in a cooperative (funded by specific mortgage schemes and/or own resources) and/or pay rent to the cooperative;
- bond finance (corporate bonds; national, regional or local bonds).

In recent decades, direct funding through state grants has decreased significantly. Social housing providers have become increasingly dependent on income from rent and sales to finance the development and management of their housing stock. This has also increased interest in attracting funding from institutional investors, as can be seen in the examples from France, the Netherlands and the UK, discussed in the introduction to this chapter. Moreover, the share of household finance has increased, most notably visible with the rise of initiatives from and/or with households to develop and manage housing. The decreased state finance and associated alternative finance sources lead to less straightforward public governance arrangements of housing and contribute to the hybridity in housing provision and governance [2].

### 2.4. Hybridity in Housing

Mullins, Czischke and Van Bortel [39] have explored hybridity in the context of social housing organizations. They refer to Anheier's [40] view that the presence of relatively persistent multiple stakeholder configurations are a necessary condition of hybridity, and to Billis' [41] view that hybrid organizations possess "significant" characteristics of more than one sector (public, private and third). Furthermore, they discuss financial dependencies hybrids (mixing state and market funding), governance structure hybrids (reflecting stakeholder mixes or separating charitable and commercial activities) and products and services hybrids (combining housing with social and neighborhood support services). A key characteristic, according to Blessing ([42], p. 190), is that hybridity implies "spanning state and market, combining public and private action logics, and [being] subject to multiple sets of institutional conditions". Therefore, Mullins et al. ([39], p. 411) stress that "one of the most compelling reasons for considering change in housing organizations through the lens of hybridity and social enterprise is to capture the underlying tensions associated with competing institutional logics" [43–45].

While Mullins et al. [39] focused mainly on hybridity in a single organization, this project takes a slightly different approach, focusing on collaborative structures between different types of actors (with varying degrees of "dominance" among actors) and with housing provision as the outcome or product of these collaborative structures. In order to classify these hybrid structures, we also need to classify the individual actors. In doing so—following Czischke, Gruis and Mullins [46]—we employ the model developed by Brandsen, Van den Donk and Putters [47], based on Pestoff [48] and Zijderveld [49], to classify organizations (see Figure 2). The authors distinguish four types of organizations, including the state, markets, communities and third-sector organizations. Supporting Buckingham [50], Czischke et al. ([46], p. 42) contend that "one limitation of this model may be that the third sector is not seen

as a domain in its own right but rather as a tension field between the state, market and community".
Following Billis [41], we view third-sector organizations as an organizational form in its own right,
acknowledging they are subject to state, market and community drivers or values. However, they
still have their own characteristics, which distinguish them from "pure" state, market or community
organizations, perhaps most clearly seen in the absence of (direct) governance through members,
shareholders or public administration.

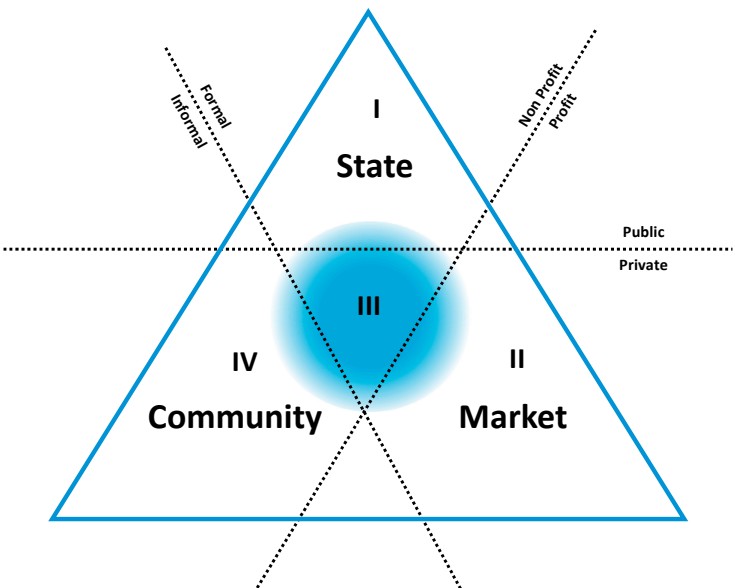

**Figure 2.** Classification of state, market, community and third-sector organizations. Sources: [47–49],
adapted by [51] (p. 50).

### 3. Austria: Increased Involvement of Market Actors through a New Financing Vehicle for Affordable Housing

Mundt and Amann [52] discuss the so-called Wohnbauinitiative (WBI) of Vienna, which is a
subsidy scheme by the municipality of Vienna providing medium-term, low-interest loans and/or
cheap building land for housing construction. It encourages new construction in the mid-price range,
granting financial benefits in exchange for limited-term social obligations by the developers concerning
rent levels and access criteria, thus adding an instrument for the affordable (non-traditional social
housing) segment.

The governance is based on the participating consortia of developers meeting certain requirements
in return for privileges. These requirements concern the minimum amount of equity to be brought
in by the consortia, the minimum investment per completed dwelling and limitations on rent levels
and capital contributions by tenants during the duration of the loan term (10 years). Furthermore,
the allocation of 50% of all completed dwellings, and 50% of all re-allocations of dwellings during
the duration of the loan term, are conducted by the general allocation agency for subsidized housing,
which allocates dwellings according to social criteria and general waiting lists, (although, in contrast to
other subsidy schemes, there are no income limits). All rental contracts have to be of an unlimited term,
and rental rates will only increase during tenancy with the overall inflation rate. Rents can be raised
when tenants relocate, and dwellings can also be offered for sale at these turnover moments. Finally,
the city's influence on the quality of submitted projects is secured through a special advisory board that
evaluates the projects and proposes improvements, much like in the other subsidy programs. These
advisory boards often include specialists from various domains, including representatives from Vienna
city council departments and independent professionals such as architects, construction specialists,
ecologists and urban planners [53].

Discussion of the WBI reveals how it emphasizes new elements within subsidy arrangements, including the orientation on both commercial and limited-profit housing providers, the inclusion of capital from institutional investors, municipal building plots and municipal medium-term loans, and the limited-term nature of maximum rents for new contracts. The initial WBI program was set up for some 6300 new dwellings to be built between 2012 and 2015 following this scheme. Due to good experience with the new subsidy scheme, a second wave of the WBI was initiated in 2015 [52].

Mundt and Amann [52] also point out some drawbacks and risks. While limited-profit housing companies have to stick to cost-rents throughout the existence of buildings, commercial developers will be allowed to raise the rent level after 10 years (for new tenants only) to a possibly higher market level, which generates insecurity about affordability on a longer term. Social targeting is not as strong as in other schemes, since there are no formal income limits, and the often-used obligation of using the home as the main residence is not a precondition in this scheme. The financial benefit of low-interest loans is dependent on the development of market interest rates, and therefore public building plot reserves were a crucial additional element of the WBI's success.

A main risk for the city of Vienna lies in paying back loans channeled to housing developers, but this is mitigated by directing the call at consortia so that the risk is spread across more developers and financial institutions, often including limited-profit housing companies with very high credit ratings. Furthermore, the city's financial involvement through loans is less pronounced in cases where discounted land for building has been provided by the city itself. Finally, the completed dwellings are usually cheaper than the market, and allocation and the marketing of the dwellings is not an issue in the current situation, which reduces the risk of vacancies [52].

## 4. Italy: Increased Involvement of Communities through Combining Co-Housing and Cooperative Housing in Affordable Housing Developments and Management

Housing for low-income households in Italy was traditionally provided by regional and local government agencies, and supported by subsidies from the national government. After the abolishment of this funding by the end of the 1990s, the production of new low-income housing plummeted. In 2008, the Italian government introduced an Integrated System of Housing Funds (Sistema Integrato dei Fondi (SIF)) to support "social housing" as a new form of affordable rental housing tenure in the Italian context. While low-income housing in Italy was traditionally provided by public actors, the SIF initiative aimed to boost social housing provided by partnerships that included a mix of not-for-profit, private, community and government actors. Ferri, Pogliani & Rizzica [53] discuss two projects supported by SIF-funding, and in particular investigate the advisory role of the Fondazione Housing Sociale (FHS)—a private, non-profit entity that promotes and carries out social and collaborative housing projects and supports the creation of partnerships to deliver this innovative type of housing. This is an example of how a non-profit organization can involve and empower residents in housing development and management (the community corner of the triangle in Figure 2), while institutional actors retain strategic control (the state, third-sector and market corners in Figure 2).

The explicit goal of the initiatives investigated by Ferri et al. [53] was not only to develop affordable housing for lower- and middle-income groups, but also to explicitly create and experiment with participatory mechanisms for community cohesion in order to foster a sense of belonging to a place of residence. These participatory mechanisms are captured in social design specifications (e.g., a carefully designed mixed tenant population), new types of "social management" (e.g., onboarding activities to support community cohesion) and innovations of participation and social accountability processes (e.g., reciprocity-based time banking systems where work hours are used as a community currency) [53].

SIF-funding has created a new source for affordable housing investments. It combines investment mechanisms developed and tested in the commercial real-estate sector, complemented by innovative collaborative forms of housing management. The Integrated System of Housing Funds, in total, comprises an amount of €2.28 billion, €1 billion of which has been invested by the Cassa depositi e prestiti (a bank closely related to the Italian state), including €140 million by the Ministry of

Infrastructure and Transport, and €888 million by banks, insurance companies and pension funds. Since 2008, the Italian social housing sector has grown to the point whereby it is now seen as an attractive sector to invest in. There are 27 approved local funds spread throughout Italy, comprising about 220 projects, 14,800 social homes and 6500 beds in temporary or student residences. The fund aims to support the construction of about 20,000 apartments.

There are also some drawbacks. The projects mostly focus on lower-middle income households. The capacity to accommodate for the most vulnerable low-income households is limited, because this could make business unfeasible. In addition, the partnerships needed to develop affordable housing projects are complex, and involve many government, market, non-profit actors and residents. Expertise, such as that offered by the Fondazione Housing Sociale, is needed to set-up these partnerships.

The case study authors [53] state that the affordable housing projects discussed constitute a combination of bottom-up resident involvement, as can be found in bottom-up co-housing initiatives [6] and housing co-operatives with a more top-down approach to resident participation. The involvement in the delivery of housing management services is more similar to the co-housing option, while asset development and management strategies are more top-down oriented.

The affordable housing projects discussed by Ferri et al. [53] mitigate risks in several ways. These projects aim to reduce housing management risks that could lead to property depreciation or lower the return on investment (e.g., rent arrears, costs associated with high turnover rates or anti-social behaviors). These risks are addressed by creating a socially and economically mixed resident population, and by involving residents in the design and construction of the projects, thus increasing community cohesion.

## 5. England: Access to Private Finance for Affordable Housing through Collaboration between Housing Associations and Local Authorities

Governments in many countries have retreated from direct involvement in the delivery of affordable housing. In England, council housing has long been a dormant legacy form of social housing, gradually declining in numbers due to the "right to buy" and housing stock transfers to housing associations. That situation has recently changed considerably, as explored by Morrison [54].

Increasingly local authorities across the UK are looking for innovative models facilitating them to take the lead again in delivering affordable homes. Moreover, the national government acknowledges that local authorities should be part of the solution to help fix the "broken" English housing market, and has begun to loosen regulatory restrictions in order to allow entrepreneurial local authorities to innovate and explore different delivery models. There has been a recent surge in local authorities' interest in setting up joint ventures with non-profit housing associations, particularly as they are able to access sources of private funding that are not available to local authorities. In her case study, Morrison explores how the UK's government regulatory framework has changed, and investigates how these innovative joint ventures between local authorities and housing associations work.

Local authority Brighton & Hove has taken the lead and established a first-of-its-kind innovative joint venture with a housing association, Hyde Housing Group, to provide 1000 affordable homes [55]. Housing developments will be genuinely affordable for low-income households, as rent are set according to national living wages. This limited liability partnership (LLP) innovatively pools resources and shares costs, risks and financial rewards.

Other local authorities are keen to learn how to set up similar joint ventures with active housing associations [56]. Specific LLPs can be set up to acquire, fund, develop and own discounted rental and shared-ownership homes, which reflect the local authorities' different circumstances and priorities. There is considerable potential to translate this innovative joint venture funding model into the European context. For this innovation to take place, a strong political and corporate leadership within the local authority is necessary [57]. Moreover, it also takes strong government backing for local authorities to take back the lead in building homes again [58].

There are large numbers of innovative variations in the use of joint ventures that can be further explored. These include not just joint ventures between local authorities and single housing associations, but also the participation of multiple housing associations. Given the potential appetite from institutional investors, these types of joint ventures can bring in equity investors at the outset or at a later date. The benefits to all parties are strong [59].

This joint-venture model of development enables local authorities to address local housing needs much more effectively. It also allows them to secure value increases of land for future reinvestment [55]. Operating through a special purpose vehicle also provides an effective means for both local authorities and housing associations to set their own rents, thus critically challenging government definitions of affordability. They also become less beholden to government restrictions on their activities. As public and non-profit organizations work more closely together across each country, they in turn take control of the pace of new housing developments. They become less affected by private, profit-driven, house-building industry practices. By doing so, they start to play a greater role in scaling up affordable housing delivery and helping to fix "broken" housing markets witnessed across Europe [54].

## 6. Discussion and Conclusions

The main driver underpinning this article was the increasing hybridity and variety of actors involved in affordable housing governance, development, management and finance. This contribution focused in particular on three cases that represent innovative arrangements to involving private actors and finance in the provision of affordable housing. In this final section, we discuss the findings from these three cases according to the two main themes of the article: how private actors are stimulated to participate, and how public housing values are secured by these arrangements.

In the selected case studies, we recognize the increased hybridity in several ways. In the case of the Wohnbauinitiative in Vienna, hybridity is actively stimulated by the local government program to provide cheap loans and land to consortia of private and third-sector actors. In Italy, resident involvement in housing development and management (creating a "community of inhabitants") is used to increase social cohesion and strengthen the rental housing sector. In addition, the Italian case demonstrates how investment funds, developed in the commercial real-estate sector, can be used to provide social housing. The English case demonstrates how joint-venture hybrids between housing associations and local authorities can create new opportunities for affordable housing delivery, whilst also opening up opportunities for private-sector investments.

Whilst the cases are chosen as examples of increased involvement of private actors in affordable housing, the cases also make clear that there is still an important role of governments in shaping a framework for affordable housing to be sustained and flourish. All three cases illustrate that innovations in affordable housing need government support. The Wohnbauinitiative in Vienna is a government initiative underpinned by government-owned land and low-interest government loans. In Italy, the social housing investments are in part supported by government contributions. The English case demonstrates how local governments can develop strategic alliances with non-profit and private actors to provide affordable housing.

The cases discussed in this article show that affordable housing innovations can be integrated into national and local housing and urban planning policies. Governments, on various hierarchical levels, can create favorable conditions for the creation of affordable housing projects by: modifying legislation and regulation; revising urban planning instruments; acting as initiators, network managers and housing market regulators; improving urban infrastructure; and devising strategies to provide land. Land allocation models based on competitive tenders can be applied in other countries where the local governments have a strong mandate to develop urban and housing policies.

The hybrid forms discussed in this article can have several benefits. They are often less dependent (costly) on state financial support, which is in a way also logical if they are aimed at the lower middle-income households and not at the core, traditional, target group of social housing. There

are also other benefits as well, including the spread of financial risks, tenant empowerment and community development.

Nevertheless, there are risks, particularly to the safeguarding of public housing values such as quality, affordability and allocation to specified target groups. Although governance arrangements mitigate risks on the shorter term, it seems difficult to secure the availability and affordability of housing, as well as maintaining an increased, active involvement of community actors, tenants and market actors, in the long run.

Some innovations might altogether challenge the notion of public housing values. The English case, for example, shows that operating through a special purpose vehicle that is not yet part of national government regulation can provide policy freedom for both local authorities and housing associations to determine housing values, specifically by establishing local standards for affordable housing that challenge national government definitions of affordability (i.e., using the national living wage standard for affordability instead of the 80% of market rent standard used by the national government).

Many of the innovations explored in this article and other studies are quite recent, and their impact on the availability of affordable housing are (still) limited or gradually emerging. More research is needed to explore the transferability of these practices to other contexts. This provides opportunities for experimentation and intensive collective learning experiences. Additional research also needs to assess the long-term impacts, and increase our understanding, of the critical success factors of the partnership hybrids underpinning these innovations. What are the enablers and barriers? Which incentives and coercive mechanisms are successful in securing the long-term involvement of private-sector actors? Finally, there is a need for further (international) comparative studies to explore and identify the transferability potential of these results, and to assess the costs and benefits of these new types of housing delivery models in greater depth.

**Author Contributions:** This article was co-created by G.v.B. and V.G.

**Funding:** This research received no external funding.

**Acknowledgments:** The three case studies explored in the article are the outcomes of a collaborative project of Delft University of technology, several members of the Working Group Social Housing: Organizations, Institutions and Governance of the European Network for Housing Research (ENHR), and several members of the European Federation for Living (EFL). We want to thank all involved for their contribution.

**Conflicts of Interest:** The authors declare no conflict of interest.

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
