# Peer review of "Innovative Arrangements between Public and Private Actors in Affordable Housing Provision: Examples from Austria, England and Italy"

_urbansci, doi:10.3390/urbansci3020052_

Round 1

Reviewer 1 Report

This is paper is a fairly straightforward discussion of examples of how government and other actors are working together to provide affordable housing. The paper notes that governments are increasingly partnering with nonprofit and third sector organizations, as well as communities. Examples from the three countries in the title of the paper are discussed.

I do not think that this paper brings anything new to the literature on affordable housing. These trends are well recognized and have been developing for decades. Also the case studies seem to be drawn more or less directly from other publications.

Author Response

See attached file for our comments.

Reviewer 2 Report

Line 147-150: the format of the table makes it difficult to interpret, please revise

Line 235: great visual!

Line 256: does "turnover" mean from tenant A to tenant B, or does it refer to the change after the 10 year period? please specify.

Line 257: Please provide examples of who serves on the "special advisory board"

Line 272: Please define "main residence" as is used in this sentence.

Line 289: Tupo "traditionally providedD by public..."

Line 302-304: please provide an example of each of the participatory mechanisms listed.

Line 305: Please use the acronym "SIF" for consistency.

Line 321: Which authors are you referring to here?

Line 329: Please define and provide examples of what 'community profile' is/looks like.

Line 330-331: Please, re-phrase to say: "mitigate the risk of property depreciation" 

Line 340: remove the word "and" or re-phrase the sentence. Remove the word "new" (since innovative implies new)

Line 343: Who is "it"? Please specify in the sentence.

Line 349: Typo, remove the s after works, for subject-verb agreement.

Line 362-366: This entire paragraph needs to be re-written for clarity, as it stands the paragraph is difficult to follow.

Line 368: Remove "Oxley et al., 2015" since you are using a different citation style throughout

Line 425: how were the national government definitions of affordability challenged? Were they raised, lowered, relaxed...? Please specify.

Author Response

See notes to Reviewer 2 in the attached document.

Reviewer 3 Report

General comments

The paper is very interesting and in general the different sections have been well structured.

Specific comments for sections

·         The Introduction section introduces the objectives well. It is advisable to clarify what the authors mean by "institutional investors" at line 38.

·         The "2. Key concepts" section the salient aspects of the issue, i.e. Affordable housing (2.1 sub-section), Housing governance (2.2. Sub-section), Housing finance (2.3.sub-section) and Hybridity in housing (2.4. sub-section).

·         In the sub-section 2.1Affordable housing the Figure 1 isn’t clear. The authors should clarify what it refers to:

“Rents in the social housing

segment are usually related to

and

“Rents in the affordable housing are

usually derived from, but lower than”.

·         Sections:

“3. Austria: increased involvement of market actors through a new financing vehicle for

 affordable housing”;

“4. Italy: increased involvement of communities through combining co-housing and housing cooperative in affordable housing development and management”;

“5. England: access to private finance for affordable housing through collaboration between housing associations and local authorities;

they are well structured, but could be enhanced.

In fact, the authors could report a flowchart or a table for each sub-section, highlighting the ways in which the issues related to the themes introduced in the “2. Key concepts in the three case studies were addressed.

In the sub-section “3. Austria: increased involvement of market actors through a new financing vehicle for affordable housing,” at line 263 what does it mean “ ...”near “contracts”.

·         The Discussion and Conclusion Section proposes a summary of the results and introduces an interesting debate on the possible repercussions of this study, coherently with the objectives indicated in the Introduction section.

Author Response

See notes to Reviewer 3 in the attached document.
